# NLRC3 Attenuates Antiviral Innate Immune Response by Targeting IRF7 in Grass Carp (*Ctenopharyngodon idelus*)

**DOI:** 10.3390/ijms26020840

**Published:** 2025-01-20

**Authors:** Lei Zhang, Haitai Chen, Xiang Zhao, Youcheng Chen, Shenpeng Li, Tiaoyi Xiao, Shuting Xiong

**Affiliations:** 1Fisheries College, Hunan Agricultural University, Changsha 410128, China; lei.zhang@stu.hunau.edu.cn (L.Z.); 15109826557@163.com (H.C.); zhaoxiang@hunau.edu.cn (X.Z.); 2Hunan Engineering Technology Research Center of Featured Aquatic Resources Utilization, Hunan Agricultural University, Changsha 410128, China; cyc20002023@163.com (Y.C.); z13647195426@163.com (S.L.); 3Yuelushan Laboratory, Changsha 410128, China

**Keywords:** *Ctenopharyngodon idella*, NLRC3, IFN response, IRF7, antiviral immunity, negative regulator

## Abstract

*NLRC3* belongs to the NOD-like receptor family and is recognized as a modulator of innate immune mechanisms. In this study, we firstly report that *Ctenopharyngodon idelus NLRC3* (*CiNLRC3*) acts as a negative regulator in the antiviral immune response. *Cinlrc3* is ubiquitously expressed across tested tissues, displaying particularly high expression in the intestine, spleen, gill and kidney. Notably, *Cinlrc3* expression is markedly upregulated following grass carp reovirus (GCRV) infection both in vivo and in vitro. Functional assays reveal that the overexpression of CiNLRC3 hampers cellular antiviral responses, thereby facilitating viral replication. Conversely, the silencing of CiNLRC3 through siRNA transfection enhances these antiviral activities. Additionally, CiNLRC3 substantially diminishes the retinoic acid-inducible gene I (RIG-I)-like receptor (RLR)-mediated interferon (IFN) response in fish. Subsequent molecular investigations indicates that CiNLRC3 interacts with the RLR molecule node, IRF7 but not IRF3, by degrading the IRF7 protein in a proteasome-dependent manner. Furthermore, CiNLRC3 co-localizes with CiIRF7 in the cytoplasm and impedes the IRF7-induced IFN response, resulting in impairing IRF7-mediated antiviral immunity. Summarily, these findings underscore the critical inhibitory role of teleost NLRC3 in innate immunity, offering new perspectives on its regulatory functions and potential as a target for resistant breeding in fish.

## 1. Introduction

In vertebrates, innate immunity serves as the first line of defense against invading pathogens, including viruses, and plays a critical role in immediate and non-specific response to infections. Type I interferons (IFNs) are pivotal in this defense mechanism, whose production initiates a cascade of signaling events that lead to the expression of numerous interferon-stimulated genes (ISGs), effectively helping the host to establish an antiviral state [1,2]. Pattern recognition receptors (PRRs), including Toll-like receptors (TLRs), retinoic acid-inducible gene I (RIG-I)-like receptors (RLRs), nucleotide-binding domain and leucine-rich repeat-containing proteins or NOD-like receptors (NLRs), absent in melanoma 2 (AIM2)-like receptors (ALRs) and C-type lectin receptors (CLRs), can accurately identify the viral pathogen-associated molecular patterns (PAMPs) and trigger the downstream immunomodulatory cascade [3], resulting in the production of type I IFNs, pro-inflammatory cytokines, and chemokines to inhibit virus replication [4,5,6,7].

Among these PRRs, NLRs are a significant group of cytoplasmic receptors that play diverse roles in regulating immune responses and maintaining cellular homeostasis. Structurally, NLRs consist of three specific domains, including an N-terminal effector domain (EBD), a central nucleotide-binding oligomerization domain (NED or NACHT), and a C-terminal leucine-rich repeat domain (LRR) [8,9]. The NACHT and LRR domains are highly conserved, while the N-terminal EBD are less conserved among different species [10]. In vertebrates, due to the N-terminal EBD possibly being a caspase-recruitment domain (CARD), a pyrin domain (PYD), a baculovirus inhibitor of the apoptosis domain (BIR), or an activation domain (AD) [11], NLRs are classified into four subfamilies, namely NLRA (AD), NLRB (BIR), NLRC (CARD) and NLRP (PYD) [12], reflecting their functional and structural diversities.

NLRC3, an NLRC group member, consists of an N-terminal caspase activation and recruitment domain, and it is conserved across the teleost [13]. NLRC3 is considered a regulatory NLR and is involved in modulating mitogen-activated protein kinase (MAPK), NF-κB, autophagy, and IFN-1 signaling pathways [14]. NLRC3 is also generally considered as an immune modulator that can inhibit signaling pathways associated with inflammation and antiviral responses [15]. In mammals, NLRC3 has been implicated in dampening the activity of pathways triggered by TLRs and RLRs, suggesting its role in preventing excessive immune activation [16]. NLRC3 is also associated with stimulator of interferon genes (STINGs) and TANK binding kinase 1 (TBK1) and blocks STING-TBK1 interaction and downstream IFN-I production, demonstrating the crossover of two key pathways of NLR and STING in innate immune regulation [17]. The expression of NLRC3 in macrophages negatively affects glycolysis and the immune defense mechanisms of the host by modulating the NF-kappaB-NFAT5 complex [18]. Additionally, NLRC3 diminishes antiviral immunity and triggers inflammatory responses in the primary brain cells of groupers after infection with nervous necrosis virus [19].

Despite the well-characterized function of NLRC3 in mammalian systems, there are few studies on how NLRC3 recognizes the virus and regulates IFN signaling in the innate immune antiviral response. Moreover, the impact of NLRC3 on the modulation of IFN-I pathways during viral infections has not been thoroughly explored in aquatic organisms. At present, grass carp (*Ctenopharyngodon idelus*) is the most productive economic fish in the world while suffering from the hemorragic disease caused by grass carp reovirus (GCRV) infection, which has long-termly restricted the healthy development of the aquaculture industry. New materials of grass carp with high disease resistance are the most fundamental solutions to fish virus diseases, but the premise of this approach is to deeply analyze the antiviral immune response mechanism. Currently, we firstly identified NLRC3 from *C. idelus* and delineated its regulatory mechanism in the antiviral immune response. Our findings have revealed a novel regulatory mechanism involved in IFN signaling, providing new insights into the key regulatory mechanisms of NLRC3 in the teleost innate antiviral immunity and new aspects of fish immunology, which may have significant implications for aquaculture and fish breeding.

## 2. Results

### 2.1. Identification of NLRC3 in Grass Carp

We searched the NLRC3 cDNA sequence from a grass carp genome database (http://bioinfo.ihb.ac.cn/gcgd/php/index.php (accessed on 20 May 2020) and cloned it. As shown, the open reading frame (ORF) of CiNLRC3 is 3417bp, encoding a protein with 1138 amino acids, consisting of a nucleotide-binding and oligomerization domain (NACHT) and several leucine-rich repeat (LRR) domains (Figure 1A). Multiple sequence alignments of the NLRC3 protein from grass carp, blunt snout bream, zebrafish, common carp, human and mouse were performed. As illustrated, the NLRC3 homology among fish is as high as 86.91–91.83%, while the homology between fish and mammals is 32.12–34.27% (Figure 1B), suggesting that NLRC3 might be functionally conserved in fish, but not across the teleost and mammals.

Subsequently, NLRC3 homologs of vertebrate species, including mammals, avian, reptilian, amphibian, teleost and cartilaginous fish, were searched from NCBI following phylogenetic tree construction. The results showed that NLRC3 homologs can be divided into three big paralleling clades. Avian and cartilaginous fish NLRC3 proteins first aggregate into a cluster and then parallel reptilian ones, forming a large clade; mammal NLRC3 proteins gather alone to form the second large clade; but teleost and amphibian NLRC3 proteins are parallel, clustering together to form the third large branch (Figure 2).

The genetic relationship of NLRC3 of grass carp was mostly close to that of *Megalobrama amblycephala*, forming a small cluster, parallel to the cluster of *Cyprinus_carpio* and *Carassius_gibelio* NLRC3 proteins. Later, together with zebrafish, it forms a Cyprinidae fish cluster. Then, it parallels with the cluster of Perciformes fish (*Oryzias_latipes*, *Oreochromis_niloticus*, *Lates_calcarifer*, *Larimichthys_crocea*, and *Siniperca chuatsi*) and Anguilliformes (*Anguilla rostrata*), respectively. Finally, they converge to form a major branch of the teleost (Figure 2).

### 2.2. Cinlrc3 Is Induced by GCRV Infection

Tissue distribution analysis showed that *Cinlrc3* was widely expressed in all examined tissues, with the highest expression in the intestine, followed by the spleen, gill, kidney, heart, liver, head kidney, and brain, and the lowest expression in the skin (Figure 3A). As for immunofluorescence, red fluorescence was only detected in the cytoplasm when DAPI was used for the nuclear staining (Figure 3B).

To investigate the function of CiNLRC3 in antiviral response, dynamic changes in *Cinlrc3* mRNA were detected after a viral challenge in vivo and in vitro. In CIKs, GCRV significantly induced *ifn* (the marker gene for IFN response) (Figure 3C). At the same time, *Cinlrc3* was also significantly induced and reached its peak at 12 h post GCRV challenge (hpc) and returned to the level before the challenge (Figure 3D). In grass carp individuals, a significant increase of *vp7*, a GCRV gene, was detected in the liver (Figure 3E), spleen (Figure 3F), kidney (Figure 3G), and gill (Figure 3H), confirming that the immersion GCRV challenge was effective. Consistently, the mRNA level of *isg15*, a typical ISG gene, was also significantly upregulated and reached the peak at 1 day in the liver, spleen and kidney (Figure 3I,K) or 3 days (in gill, Figure 3L) post GCRV infection (dpi) and then returned to the initial level, indicating that the challenge experiment successfully induced an IFN response. Meanwhile, *Cinlrc3* mRNA was also significantly induced after GCRV infection in the liver, spleen and kidney, similar to *ifn* and *isg15* (Figure 3M,P). Taken together, *Cinlrc3* was remarkably upregulated by GCRV in vivo and in vitro, suggesting that *Cinlrc3* might participate in the antiviral immune response.

### 2.3. CiNLRC3 Dampens the Cellular Antiviral Response

To further investigate the role of CiNLRC3 in viral replication, CIK cells were overexpressed with CiNLRC3, followed by GCRV infection. As illustrated in Figure 4A, the CiNLRC3-transfected CIK cells exhibited more severe cytopathic effects (CPEs), suggesting that the overexpression of NLRC3 accelerated cellular viral replication. Meanwhile, compared with the control cells transfected with an EV (empty vector), the CIK cells transfected with CiNLRC3 showed a significant upregulation of *Cinlrc3*, confirming the reliability of the transfection experiment (Figure 4B). Under the same conditions, two GCRV genes, *vp4* and *vp5*, transcriptionally increased remarkably in CiNLRC3-overexpressed cells (Figure 4C,D), which was consistent with the results of the above crystal violet assay. On a molecular level, the overexpression of CiNLRC3 significantly attenuated the transcription level of cellular *ifn*, which was the most crucial antiviral factor in the IFN response (Figure 4E).

To further investigate the function of endogenous CiNLRC3 in viral replication, two CiNLRC3-specific siRNAs were designed. In the CiNLRC3 knockdown efficiency assay, CiNLRC3#siRNA-1 and CiNLRC3#siRNA-2 significantly knocked down the expression of endogenous CiNLRC3 when compared with the negative control group (siRNA-NC) (Figure 4F). In the subsequent qPCR assays, *vp4* and *vp5* displayed a significant reduction in CiNLRC3#siRNA-1/2-tranfected cells (Figure 4G,H) and ISG genes, *mx* and *viperin*, were significantly upregulated after CiNLRC3 knockdown (Figure 4I,J). These data demonstrate that CiNLRC3 suppresses the cellular antiviral response.

### 2.4. CiNLRC3 Blocks the RLR-Mediated IFN Response

How does CiNLRC3 function in antiviral response? As known, IFN response is proviral in antiviral response, and we explored the role of CiNLRC3 in IFN response. As shown, the activation of the fish IFN promoter was significantly induced by poly I:C, GCRV and SVCV stimulation, while the overexpression of CiNLRC3 remarkably inhibited this induction of IFN (Figure 5A).

RLR-mediated-type I IFN production is critical in innate antiviral immune process. Does CiNLRC3 negatively impact fish RLR-mediated IFN response? As illustrated in Figure 4B, IFNφ1 promoter activity is significantly induced by the overexpression of each RLR signaling molecule, including RIG-I, MDA5, MAVS, MITA, TBK1, IRF3 and IRF7. However, this activation is inhibited by the co-transfection of CiNLRC3 (Figure 5B), verifying the hypothesis that CiNLRC3 negatively impacts IFN response through RLR signaling.

### 2.5. NLRC3 Interacts with IRF7 and Degrades IRF7in a Proteasome-Dependent Manner

Which molecular node in RLR signaling does CiNLRC3 interact with and function in the IFN response? First, a Co-IP experiment was performed to screen the potential molecules in RLR signaling. As demonstrated in Figure 6A, the anti-HA Ab-immunoprecipitated protein complexes are clearly recognized by the anti-Flag Ab when TBK1/IRF3/IRF7-HA is co-transfected with CiNLRC3-Flag but not EGFP/MITA -HA, suggesting that TBK1/IRF3/IRF7 associates with CiNLRC3. Also, there is a weak-signal band detected when MITA-HA is co-transfected with CiNLRC3-Flag (Figure 6A). Since IRF3 and IFR7 are the last nodes in RLR signaling, we continue to verify the association between NLRC3 and IRF3/7. Does NLRC3 have any effect on IRF3/7 at the protein level? We co-transfect the plasmid expressing Flag- NLRC3 and HA-IRF3 or HA-IRF7 into CIK cells and conduct a WB analysis. As illustrated, there is no significant differences in the level of the IRF3-HA protein when different doses of the NLRC3-Flag plasmid are co-transfected (Figure 6B), while the protein level of IRF7-HA is decreased by overexpressing NLRC3-Flag in a dose-dependent manner (Figure 6C). Furthermore, we treat cells with MG132, a proteasome inhibitor, to probe the underlying mechanism of NLRC3 on IRF7. As shown in Figure 6D, NLRC3 causes the degradation of IRF7, which can be rescued by the existence of MG132, indicating that the degradation of IRF7 by NLRC3 is proteasome-dependent.

Further confocal microscopy was used to determine the interaction of CiNLRC3 and IRF7. The transfection of NLRC3-GFP or IRF7-mCherry revealed the cytoplasmic co-localization of NLRC3 and IRF7 (Figure 6E,F) in HEK293T cells. These results indicated that CiNLRC3 interacts with and degrades IRF7 in a proteasome-dependent manner.

### 2.6. CiNLRC3 Impairs IRF7-Mediated Cellular Antiviral Response

Given that CiNLRC3 targets CiIRF7 for protein degradation, and fish IRF7 is considered a strong and important antiviral factor [20], we next investigated whether CiNLRC3 impaired the cellular antiviral response triggered by IRF7. Three group experiments were performed in CIK cells with an equal EV, the expression plasmid IRF7, and the expression plasmids IRF7 plus NLRC3, respectively, and GCRV was added into cells at 24 h post transfection. As illustrated in Figure 7A, the overexpression of CiIRF7 alone significantly induces the expression of cellular ISG genes, including *isg15* and *isg20*. However, CiNLRC3 overexpression significantly decreases the mRNA level of these ISG productions (Figure 7A,B). Similar assays are repeated in CIK cells, followed by GCRV infection. The qPCR assays results indicate that three GCRV genes, including *vp5*, *vp6* and *vp7*, are inhibited by the transfection of CiIRF7, but this suppression is rescued by the co-transfection of NLRC3 (Figure 7C–E). All these results suggest that NLRC3 dampens the IRF7-mediated cellular antiviral response.

## 3. Discussion

IFNs and ISGs are critical components in the host defense against viral infection, but too small or great a production of IFNs and ISGs can lead to many diseases [21]. The precise regulatory mechanism of IFN response has been extensively studied, while the regulatory role of NLRC3 on host IFN response is poorly understood in grass carp, which have long-term suffered from viral diseases. RLR-mediated IFN response is essential for the innate immune response to RNA virus [22]. Therefore, various regulatory mechanisms of RLR signaling are of great significance for immune homeostasis [23]. In this study, our findings demonstrate that grass carp NLRC3 is a negative regulator of the RLR pathway in grass carp. It interacts with IRF7 and degrades IRF7 in a proteasome-dependent manner to negatively regulate the IFN response.

For the sequence analysis, the NLRC3 homology between fish species is high but low between fish and mammals (Figure 1B). NLRC3 attenuated the IFN responses by impacting the TRAF6/NF-κB activity and exhibited reduced IFN sensitivity, promoting the nervous necrosis virus (NNV) replication kinetics in primary grouper brain cells [24]. In this study, grass carp NLRC3 is also a negative regulator of antiviral immunity. In mammals, Nlrc3^−/−^ mice infected with Hantaan virus (HTNV) showed higher viral loads in the serum, spleen, and kidney than wild-type mice [25]. The above evidence reinforces our hypothesis that NLRC3 might be functionally conserved in fish but not across the teleost and mammals.

The subsequent analysis of NLRC3 expression characteristics showed that *Cinlrc3* was widely distributed in tested tissues, especially high in the intestine, spleen, gill and kidney (Figure 3A). While in Asian sea bass, a kind of sea fish, high expression was observed in the gill, hindgut, and midgut, followed by the truck kidney, liver, spleen, and head kidney [26]. Therefore, NLRC3 is indeed highly expressed in immune-related tissues, and the different expression distribution in various species might by related to their divergent survival environments [27]. In grouper cells, *nlrc3* displayed a significant upregulation in the brain, gill, head kidney and spleen following NNV infection [19]. Also, grass carp *nlrc3* was significantly induced both in vivo and in vitro after the GCRV challenge. Moreover, whether the NLRC3 promoter could be induced by IFN or IFN stimulation is also an interesting question to determine whether NLRC3 is an ISG gene. Altogether, whether NLRC3 is an ISG or not, there is no doubt that fish NLRC3 is involved in the antiviral immune response.

Viral diseases are a challenge for aquaculture. How does NLRC3 function in the antiviral immune process? Poly (I:C), SVCV and GCRV could efficiently trigger fish RLR signaling and induce IFN production [28]. As known, IRF3 and IRF7 are key node molecules in RLR signaling, playing an essential role in the virus-triggered induction of type I IFNs, and are also important transcription factors in regulating the expression of IFN-I and ISGs [24,29,30]. In the current study, NLRC3 was revealed to be a negative regulator of the RLR-triggered IFN response, as grass carp IFN promoter activations by all RLR signaling molecules (RIG-I, MDA5, MITA, TBK1, IRF3 and IRF7) were severely inhibited by CiNLRC3. NLRC3 could interact with MITA, TBK1, IRF3 and IRF7 in the co-IP experiment (Figure 6A), suggesting that NLRC3 may exert its inhibitory effects at multiple points within the RLR signaling cascade. Interestingly, there is no band detected in the co-IP experiment between NLRC3 and IRF3 in mammals [31], indicating potential evolutionary divergence in how NLRC3 modulates the immune response across different species. Like fish NLRC3, fish IRF3 showed significant sequence divergence from mammalian orthologs [32]. Why does the interaction between NLRC3 and IRF3 differ between mammals and fish? Is this caused by the differences in the IRF3 sequence or the NLRC3 sequence between mammals and fish? This is worth further study. Subsequent studies should also explore the interaction pattern between CiNLRC3 and other signaling molecules in the RLR signaling pathway to further elucidate the regulatory network involved in CiNLRC3.

The degradation of IRF7 by NLRC3 is proteasome-dependent, while ubiquitination is an important process in the proteasome-dependent degradation process, so further ubiquitination experiments could be conducted to solve this question. Additionally, CiNLRC3 knockdowns in vitro have demonstrated that the sharp reduction in NLRC3 can effectively promote the transcription of antiviral molecules and strongly inhibit viral replication in CIK cells. Therefore, in vivo studies assessing the resistance of NLRC3-deficient fish models to virus infection will provide deeper insights into its potential utility in resistant breeding. As NLRC3 might exert other biological functions rather than have a role in IFN response [33], NLRCs plays an important role in the bacterial recognition of animals [34] and is an anti-bacteria factor in fish [35]. Moreover, zebrafish nlrc3-like is required for microglia maintenance and macrophage homeostasis in the brain [36] and nlrc3-like deficiency leads to decreased bacterial burden at an early stage after *Mycobacterium marinum* infection [37]. Therefore, further studies in fish are also encouraged to explore more mechanisms by which NLRC3 affects various aspects, such as immune regulation, growth, or development.

## 4. Materials and Methods

### 4.1. Cell Lines, Viruses, and Fish

C. idelus kidney tissue cell line (CIK) and C. idelus ovary cell line (CO) were cultured with M-199 medium (HyClone, Utah Loga, UT, USA) at 28 °C, and HEK293T cells were cultured with DMEM basic medium (HyClone, Utah Logan, UT, USA) at 37 °C in a humidified incubator with 5% CO_2_. All cells were supplemented with 10% inactivated bovine fetal bovine serum (Gibico, Miami, CA, USA) and 100 U/mL penicillin–streptomycin–gentamicin (C0223-100mL, Beyotime Biotechnology, Shanghai, China). The GCRV JX-0901 strain (used in cell lines) induces significant cytopathic effects in CIK cell lines, and the GCRV-Huan1307 strain (used in individuals) causes severe hemorrhage symptoms in individuals. All the cell lines and viruses used in this experiment were long-term kept in our laboratory. The size of grass carp used for GCRV immersion in the experiment was about 20 cm. Before infection, the fish were kept temporarily for a week in indoor circulation breeding systems at 28 °C. All the experiments involving fish handlings were approved by the Animal Care Committee of Hunan Agricultural University, China (approval number: HNAU2024182).

### 4.2. Sequence Analysis and Phylogenetic Analysis

The cDNA and protein sequences of CiNLRC3 (OR282536.1) and CiIRF7 (XM_051884531.1) were obtained from the NCBI (https://www.ncbi.nlm.nih.gov/ (accessed on 20 May 2020). Multiple alignments were performed with ClustalW using, BioEdit, version 7.2, and the main domains were analyzed in SMART (https://smart.embl.de/ (accessed on 20 May 2020). In addition, the phylogenetic tree was drawn by neighbor-joining methods in the MEGA6.0 program, and bootstrap confidence values were based on 1000 bootstrap replications.

### 4.3. RNA Extraction, cDNA Synthesis, and Real-Time qPCR

For the tissue distribution, three healthy grass carp tissues, including intestine, spleen, gill, kidney, heart, liver, head kidney, brain, and the skin, were collected, respectively. For the GCRV immersion experiment, GCRV-Huan1307 was diluted to 500 times and then a total of 100 grass carps (20 cm in average) were divided into 2 groups, with 10 grass carp in the control group and 90 grass carp in the treatment group immersed in GCRV dilution for 30 min. The liver, spleen, kidney and gill of 9 grass carps (a mixture of 3 fish samples was considered a biological replicate) were sampled from treatment group fish after immersing for 0 days (control group), 1 day, 3 days and 7 days post immersion (dpi) and frozen in liquid nitrogen following RNA isolation. At the same time, CIK cells were seeded overnight in the six-well plate and then treated with GCRV-JX0901 when the cell density was approximately 5 × 10^5^ cells/mL. Then, the cell samples were collected at 0 h, 12 h, 24 h, 36 h, and 48 h post challenge (dpc). Subsequently, total RNA extraction, the first-strand cDNA synthesis and quantitative real-time PCR were performed with the relevant kits (ET101-01-V2, TransGen Biotech, Beijing, China; K1622, Thermo, Waltham, MA, USA; EM701-01, Novozan, Nanjing, China) and all the operations were performed according to the instructions. For qPCR, all samples were analyzed in triplicate, and the expression values were normalized to β-actin. Primers used for RT-qPCR analysis are listed in Appendix A.

### 4.4. Plasmids and SiRNA

Expression plasmids CiNLRC3-Flag/HA and CiIRF7-Flag/HA were generated by inserting their coding sequences into the HindIII and BamHI site of pcDNA3.1(+)-Flag or pcDNA3.1(+)-HA vector, and CiNLRC3- pEGFP-Flag was generated by inserting its coding sequences into the BamHI site of pEGFP-N1-Flag, and CiIRF7-Cherry was generated by inserting its coding sequences into the BamHI site of pcs2 + 8 cm-Cherry, respectively. Grass carp RLR signal node plasmids and IFN promoter plasmids were kindly donated by Xiao Wuhan Lab from the Institute of Hydrobiology, the Chinese Academy of Sciences. For the knockdown experiment, two siRNAs (#1 and #2) of grass carp NLRC3 were designed. CiNLRC3-siRNA and NC were synthesized by Tsingke Biotech (Beijing, China), and the detailed sequences are listed in Appendix A. CIK cells were applied for the transfection of 100 nM CiNLRC3-siRNA#1/2 or the negative control (NC). 

### 4.5. Transfection and Luciferase Activity Assays

Transfection was performed by Polyethylenimine Linear (MW25000, Aldrich, St. Louis, MO, USA) according to the manufacturer’s protocol as previously reported [38]. For luciferase activity assays, EPC cells seeded in 48-well plates at a density of 5 × 10^5^ cells/mL were co-transfected with indicated plasmids at a ratio of 10:10:1 (expression plasmids, promoter-driven luciferase constructs, pRL-TK). Cells were collected and lysed according to the dual-luciferase reporter assay system (Promega, Madison, WI, USA) 24 h later. Luciferase activities were measured by the LuminoskanTm Ascent (2805621, Thermo, Waltham, MA, USA) and normalized to the amounts of Renilla luciferase activities as previously described [38].

### 4.6. Antiviral Activity Analysis

Antiviral experiments were conducted as described [39]. In brief, CIK cells seeded in 6-well plates overnight were transfected with CiNLRC3 or the empty control plasmid when the cell density was approximately 5 × 10^5^ cells/mL, followed by treatment with GCRV 12 h post transfection. About 48 h later, cells were fixed and stained with 0.1% crystal violet for CPE observation. Meanwhile, we performed parallel experiments and collected the cells for RT-PCR to detect the expression of ISG genes and GCRV genome genes.

### 4.7. Immunofluorescence and Confocal Microscopy Assays

CIK cells were seeded in six-well plates and transfected with the NLRC3-HA plasmid, and then the samples were collected after 24 h. The cells were washed three times, fixed with 4% paraformaldehyde for 30 min, and then washed three times with PBS again. Subsequently, 0.2% Triton X-100 was added for 15 min, and TBST was added three times to wash the cells. The cells were incubated for 1 h in 5% skim milk for blocking, and the anti-HA antibody (1:500, Rabbit mAb, cell signaling Technology, Boston, MA, USA) was added for overnight incubation at 4 °C. The cells were washed three times with TBST, and a fluorescent secondary antibody (iFluor™ 594 coupled goat anti-rabbit IgG polyclonal antibody, HuaAn Biotechnology Co., Ltd., Hanzhou, China) was added for 1 h. Then, the cells were washed three times with TBST, and DAPI (500 µg/mL) was added for 10 min. For co-location experiments, HEK293T cells were plated overnight on microscopic cover glasses in 6-well plates, which were transfected with fluorescent-labeled plasmids when the cell density was approximately 1 × 10^6^ cells/mL. About 24 h later, cells were washed three times with PBS, and 4% (*v/v*) paraformaldehyde was used to fix them for 30 min at room temperature. After washing with PBS three times again, cells were incubated with 0.2% Triton X-100 for 15 min. Finally, DAPI (500 µg/mL) was used to nuclear stain for 10 min. Cells were examined under a confocal microscope (ZEISS, Oberkochen, Germany).

### 4.8. Co-Immunoprecipitation (Co-IP) and Western Blotting (WB)

Co-immunoprecipitation (Co-IP) assays and Western blotting (WB) were performed as previously described [39]. The relevant primary antibodies were used: anti-Flag and anti-HA (Cell Signaling Technology, Boston, MA, USA) and anti-β-actin (Bio-Rad, Hercules, CA, USA).

### 4.9. Data Collection and Statistical Analysis

The data were shown as mean ± SD. The figures were drawn by GraphPad Prism 6 software. All significant differences were assessed by the *t*-test analysis in SPSS software, version 26.0. Statistical significance was represented by asterisks (* *p* < 0.05).

## 5. Conclusions

In conclusion, we firstly cloned the NLRC3 cDNA from grass carp and analyzed its expression characteristics. The functional analysis disclosed that CiNLRC3 interacted with and degraded IRF7 to suppress IFN signaling in a proteasome-dependent manner. These findings provide a novel theoretical understanding of the fish immune regulation mechanism.

## Figures and Tables

**Figure 1 ijms-26-00840-f001:**
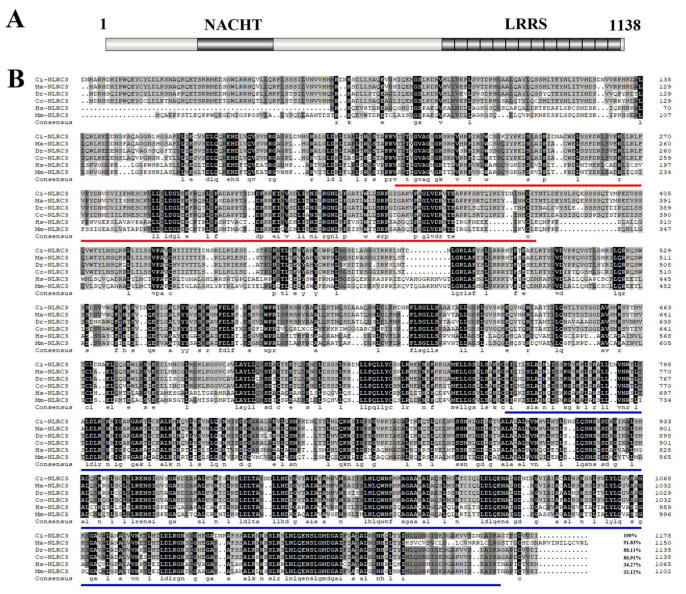
CiNLRC3 sequence analysis. (**A**) Structure illustration of CiNLRC3 protein. Structure domains were indicated in a dark frame. (**B**) Multiple alignment of NLRC3 protein sequences from grass carp (OR282536.1), blunt snout bream (XM_048174542.1), zebrafish (XM_009297629.4), common carp (XM_042752008.1), human (FJ889357.1) and mouse (XM_011245902.3). The NACHT domain is marked by red underline and the LRR domains are indicated by blue underlines.

**Figure 2 ijms-26-00840-f002:**
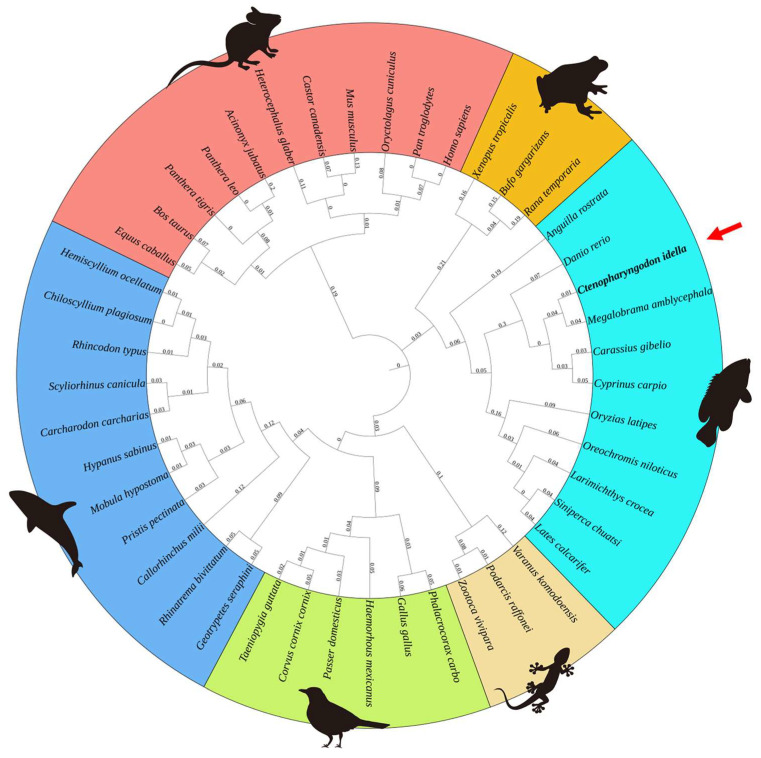
Phylogenetic tree of NLRC3 protein homologs. The phylogenetic tree was constructed using the neighbor-joining method implemented in the MEGA 6.0 software. Bootstrap confidence values, displayed at the nodes of the tree, were calculated based on 1,000 bootstrap replications. NLRC3 homologs are listed below. Mammalian: *Homo sapiens* (NP_849172.2), *Pan troglodytes* (XP_016784787.3), *Oryctolagus cuniculus* (XP_051692203.1), *Mus musculus* (NP_001074749.1), *Castor canadensis* (XP_020012107.1), *Heterocephalus glaber* (XP_004864808.1), *Acinonyx jubatus* (XP_053069323.1), *Panthera tigris* (XM_042971676.1), *Bos taurus* (XP_059737649.1), *Equus caballus* (XP_001499317.2), *Panthera leo* (XM_042921978.1); Reptilian: *Varanus komodoensis* (XP_044289647.1), *Bodarcis raffonei* (XP_053220626.1), *Zootoca vivipara* (XP_034987245.1); Avian: *Phalacrocorax carbo* (XP_064318183.1), *Gallus gallus* (XP_015150161.3), *Haemorhous mexicanus* (XP_059718048.1), *Passer domesticus* (XP_064246427.1), *Corvus cornix cornix* (XP_019136980.2), *Taeniopygia guttata* (XP_030140601.3; Amphibians: *Xenopus tropicalis* (XP_017952746.1), *Bufo gargarizans* (XP_044160788.1), *Rana temporaria* (XP_040214376.1); Cartilaginous fish: *Scyliorhinus_canicula* (XP_038676696.1), *Heterocephalus_glaber* (XP_004864808.1), *Callorhinchus_milii* (XP_007891876.1), *Castor_canadensis* (XP_020012107.1), *Rhinatrema_bivittatum* (XP_029432650.1), *Carcharodon_carcharias* (XP_041062594.1), *Mobula_hypostoma* (XP_062915561.1), *Hypanus_sabinus* (XP_059834825.1), *Pristis_pectinata* (XP_051877340.1), *Rhincodon_typus* (XP_048465466.1), *Chiloscyllium_plagiosum* (XP_043567467.1); and Teleost: *Danio_rerio* (XM_009297629.4), *Ctenopharyngodon idella* (XP_051737605.1), *Megalobrama_amblycephala* (XP_048030497.1), *Carassius_gibelio* (XP_052451976.1), *Cyprinus_carpio* (XP_042607942.1), *Oryzias_latipes* (XP_004080575.1), *Oreochromis_niloticus* (XP_003438651.1), *Lates_calcarifer* (XP_018537323.1), *Larimichthys_crocea* (XP_010730059.1), *Siniperca chuatsi* (XM_044177955.1), *Anguilla rostrata* (XP_064178397.1).

**Figure 3 ijms-26-00840-f003:**
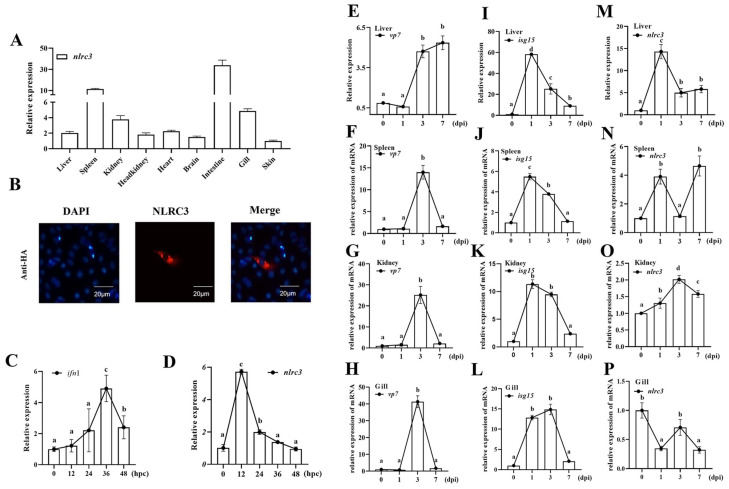
*Cinlrc3* is induced after GCRV infection. (**A**) The distribution of *Cinlrc3* in the intestine, spleen, gill, kidney, heart, liver, head kidney, brain, and skin of grass carp. (**B**) The CiNLRC3 protein is in the cytoplasm. Immunofluorescence cellular localization is performed using constructed HA-CiNLRC3. The plasmid of CiNLRC3-HA is transfected in CIK cells, and the HA antibody is utilized to detect the CiNLRC3-HA fusion protein which is indicated in red fluorescence. DAPI is used for the nuclear staining. Scale bar: 20 μm. (**C**,**D**) CIK cells are challenged with GCRV-JX0901, and cell samples are collected at 0 h, 12 h, 24 h, 36 h, and 48 h. Then, the transcriptional levels of *ifn1* (**C**) and *Cinlrc3* (**D**) are detected by qPCR. (**E**–**P**) Grass carps are immersed in GCRV-Huan1307 for 30 min and the liver, spleen, kidney and gill are sampled at 0, 1, 3 and 7 dpi. *Cinlrc3*, *vp7* and *isg15* mRNA in the liver (**E**,**I**,**M**), spleen (**F**,**J**,**N**), kidney (**G**,**K**,**O**) and gill (**H**,**L**,**P**) are detected using qPCR. Letters with the same superscript indicate no significant difference (*p* < 0.05).

**Figure 4 ijms-26-00840-f004:**
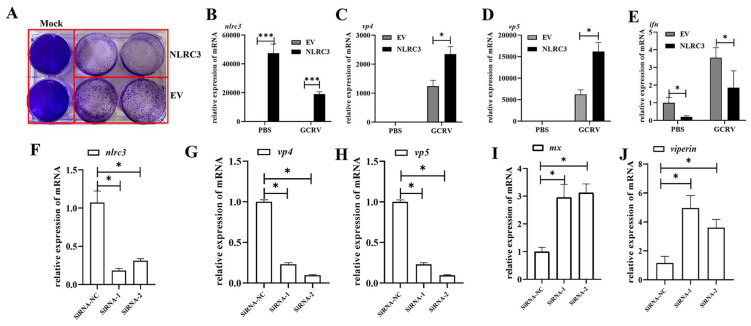
CiNLRC3 dampens the cellular antiviral response. (**A**) CIK cells were seeded into 6-well plates and transfected with an EV or CiNLRC3 (2 μg), respectively. After transfection for 24 h, GCRV was added into the transfected wells. After 36 hpi, the cells were fixed with 4% paraformaldehyde, washed three times with PBS, and then stained with 1% crystal lavender. (**B**–**E**) Under the same transfected experiments above, samples were collected at 36 hpi. qPCR was performed to detect mRNA levels of *Cinlrc3* (**B**), *vp4* (**C**), *vp5* (**D**) and *ifn1* (**E**). (**F**,**J**) CIK cells were seeded into 6-well plates and transfected with siRNA-NC or siRNA-1/2, respectively. After transfection for 12 h, GCRV was added into the transfected wells. After 36 hpi, the cells samples were collected and qPCR was applied to detect the expression of *Cinlrc3* (**F**), *vp4* (**G**), *vp5* (**H**), *mx* (**I**) and *viperin* (**J**). Asterisks indicate significant differences (* *p* < 0.05, *** *p* < 0.001).

**Figure 5 ijms-26-00840-f005:**
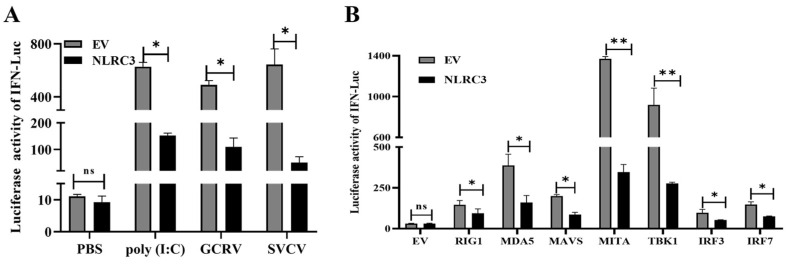
CiNLRC3 blocks the RLR-mediated IFN response. (**A**) The GCO cells are seeded in a 24-well plate overnight and then co-transfected with an EV or CiNLRC3 (500 ng), CiIFN1pro-Luc (100 ng), and pRL-TK (10 ng). Moreover, 12 h later, poly (I; C), GCRV, SVCV is added into cells, respectively. Another 24 h later, the samples are collected after 24 h following a dual-luciferase activity assay. (**B**) The GCO cells are seeded in a 24-well plate overnight and then an EV or the CiNLRC3 (200 ng) plasmid and CiIFN1pro-Luc (100ng), PRL—TK (10 ng), are transfected into the cells. At the same time, the expressing plasmids including RIG-I, MAD5, MAVS, MITA, TBK1, IRF3, and IRF7 (200 ng) are transfected into the cells, respectively. In addition, 24 h later, the samples are collected for the dual-luciferase activity assay. Asterisks indicate significant differences (* *p* < 0.05, ** *p* < 0.01).

**Figure 6 ijms-26-00840-f006:**
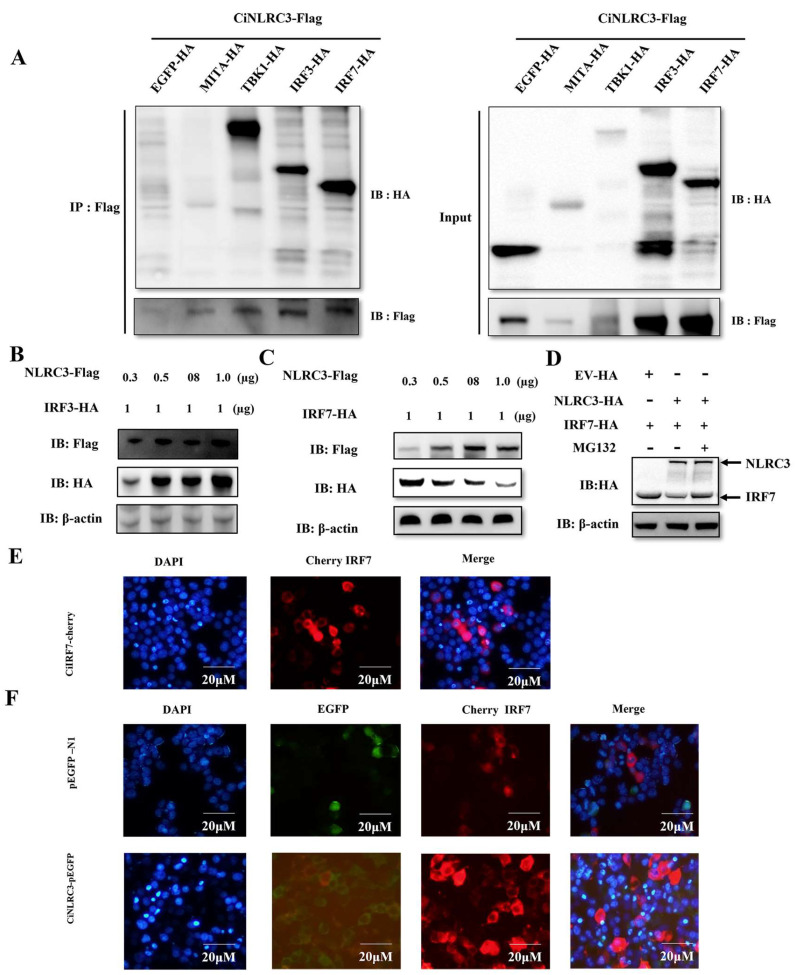
NLRC3 interacts with IRF7 and degrades IRF7 in a proteasome-dependent manner (**A**) MITA, TBK1, IRF3 and IRF7 are NLRC3-interacting proteins. GCO cells are seeded in 10 cm^2^ plates overnight, and then CiNLRC3 is co-transfected with EGFP-HA, MITA-HA, TBK1-HA, IRF3-HA, and IRF7-HA (5 μg each). After 36 h, the cells are collected for the Co-IP experiment. (**B**–**C**) NLRC3 degrades IRF7 but not IRF3. GCO cells are seeded in 6-well plates overnight, and then IRF3 (**B**) or IRF7-HA (**C**) (1 μg) and NLRC3-Flag (0.3, 0.5, 0.8, 1.0 μg) are co-transfected, respectively. After 36 h, cells are collected, and their bands are detected by Western blotting. (**D**) NLRC3 degrades IRF7 in a proteasome-dependent manner. GCO cells are seeded in three 6-well plates overnight, one plate co-transfected with IRF7-HA and an EV and the other two plates co-transfected with a repeat of IRF7-HA (1 μg) and NLRC3-HA (1.0 μg). Moreover, 24 h later, the indicated cells are treated with MG132 (20 μM) for 6 h. After 36 h, cells are collected, and their bands are detected by Western blotting. (**E**,**F**) IRF7 is co-located with NLRC3. 293T cells are seeded in 12-well plates overnight and transfected with Cherry-IRF7 (**E**), CiNLRC3-EGFP or EGFP and Cherry-IRF7 (1 μg each) (**F**) and fixed with 4% paraformaldehyde for 15 min after 24 h. Then, PBS is used for washing three times, DAPI is used for staining for 5 min, and photographs are taken under the microscope.

**Figure 7 ijms-26-00840-f007:**
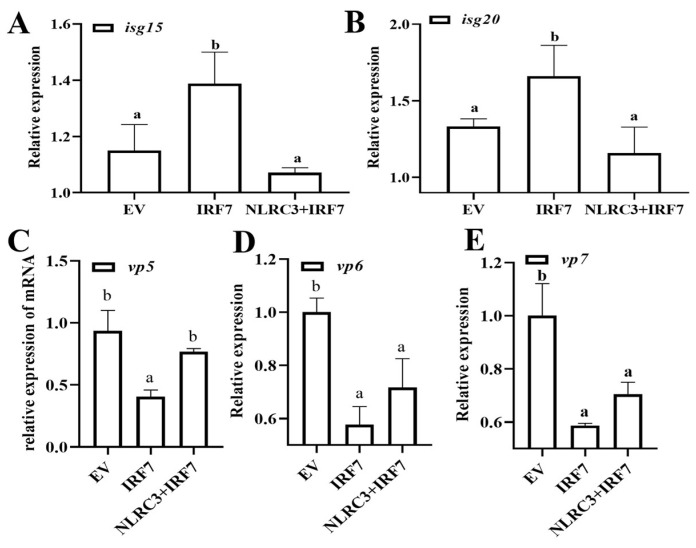
CiNLRC3 impairs IRF7-mediated cellular antiviral response. (**A**–**E**) CIK cells are seeded in six-well plates overnight, and then EV (1 μg) or EV (0.5 μg) + CiIRF7 (0.5 μg) or CiNLRC3 (0.5 μg) + CiIRF7 (0.5 μg), respectively, and GCRV are added 24 h later. The mRNA levels of isg15, isg20 (**A**,**B**), vp5, vp6 and vp7 (**C**–**E**) are detected by qPCR after another 24 h. Letters with the same superscript indicate no significant difference (*p* < 0.05).

## Data Availability

The raw data supporting the conclusions of this article will be made available by the authors on request.

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
