# Peer review of "NLRC3 Attenuates Antiviral Innate Immune Response by Targeting IRF7 in Grass Carp (Ctenopharyngodon idelus)"

_ijms, 2025, doi:10.3390/ijms26020840_

Round 1

Reviewer 1 Report

Comments and Suggestions for Authors

Present study cloned the NLRC3 gene from grass carp (CiNLRC3) and revealed its functions in antiviral innate immunity. Expression of CiNLRC3 was upregulated after grass carp reovirus (GCRV) infection. Overexpression of CiNLRC3 inhibited antiviral response and facilitated GCRV replication. It seems like CiNLRC3 interacts and prompted the degradation of key molecules of RLR pathway. There are some suggestions that needed to be considered.

1. Line 54-55, what does the species indicate?

2. The similarity between different NLRC3 homologues should be displayed in Figure 1B.

3. The resolution ratio of these fluorescence pictures is too low.

4. Line 163, the authors stated that Cinlrc3 mRNA was significantly induced by GCRV infection. But it is not in the Gill (Figure 3P).

5. Line 186, delete genome.

6. It seems like the CiNLRC3 not only interacted with IRF7 but also TBK1 and IRF3. Why IRF7 was chosen in the following experiment?

7. As a downstream molecule, IRF7 is not a exclusive molecule in RLR pathway, which is also involved in other signaling pathways. So, the conclusion that CiNLRC3 inhibited RLR-mediated IFN response through degrading IRF7 is not soundness.

Author Response

Present study cloned the NLRC3 gene from grass carp (CiNLRC3) and revealed its functions in antiviral innate immunity. Expression of CiNLRC3 was upregulated after grass carp reovirus (GCRV) infection. Overexpression of CiNLRC3 inhibited antiviral response and facilitated GCRV replication. It seems like CiNLRC3 interacts and prompted the degradation of key molecules of RLR pathway. There are some suggestions that needed to be considered.

1. Line 54-55, what does “the species” indicate?

Response: Thanks for your nice question. I apologize for the confusion caused by our improper expression. “the species” indicates different species. We have revised the manu “The NACHT and LRR domains are highly conserved while the N-terminal EBD are less conserved among different species” (Lines 54).

2. The similarity between different NLRC3 homologues should be displayed in Figure 1B.

Response: We have added as suggested.

3. The resolution ratio of these fluorescence pictures is too low.

Response: We have revised it in Fig. 3b.

4. Line 163, the authors stated that Cinlrc3 mRNA was significantly induced by GCRV infection. But it is not in the Gill (Figure 3P).

Response: Thank you for your patient review, I apologize for our inaccurate description, and I have already modified this part of the description “Meanwhile, Cinlrc3 mRNA was also significantly induced after GCRV infection in liver, spleen and kidney, similarly with ifn and isg15” (Lines 163-164).

5. Line 186, delete“genome”.

Response: We have deleted it as suggested (Line 186).

6. It seems like the CiNLRC3 not only interacted with IRF7 but also TBK1 and IRF3. Why IRF7 was chosen in the following experiment?

Response: Thanks for your comments, we have explained it in line 241-242 (Since IRF3 and IFR7 is the last node in RLR signaling, we continue to verify the association between NLRC3 and IRF3/7.) and line 245-346 (As illustrated, there is no significant differences in the level of IRF3-HA protein when different dose of NLRC3-Flag plasmid was co-transfected), so we chose irf7 for the following experiment.

7. As a downstream molecule, IRF7 is not a exclusive molecule in RLR pathway, which is also involved in other signaling pathways. So, the conclusion that CiNLRC3 inhibited RLR-mediated IFN response through degrading IRF7 is not soundness.

Response: Thanks for your doubts. We did not propose the conclusion “CiNLRC3 inhibited RLR-mediated IFN response through degrading IRF7”, we put forward that “CiNLRC3 negatively impacted IFN response through RLR signaling (which was proved by the IFN promoter activity detection [line 216-221] )” and ”NLRC3 dampens the IRF7-mediated cellular antiviral response[line 272-283]”

Reviewer 2 Report

Comments and Suggestions for Authors

In the manuscript submitted to me for review entitled "NLRC3 attenuates antiviral innate immune response by targeting IRF7 in grass carp (Ctenopharyngodon idelus) the authors Lei Zhang, Hai-tai Chen, Xiang Zhao, You-cheng Chen, Shen-peng Li, Tiao-yi Xiao and Shu-ting Xiong present a study in which they provide a new theoretical understanding of the mechanism of immune regulation in grass carp fish.

The study was conducted in vitro and in vivo, and the experiments with fish were approved by the Animal Care Committee of Hunan Agricultural University, China.

My remarks and recommendations to the authors are:

1. The Materials and Methods section does not describe how many experimental groups the fish used (grass carp) were divided into and how many specimens were included in each group.

2. On lines 384-385 it is stated:

"At the same time, CIK cells were seeded in the six-well plate and treated with GCRV-JX0901."

Was the treatment performed immediately after the cells were seeded or after the formation of the cell monolayer?

3. In the same sentence on lines 384-385, the density of the cells that were seeded is not stated. Let it be added.

The same remark applies to sections 4.6. and 4.7.

4. In section 4.7. in several places the grammatically correct tense is not used to describe the conduct of the experiment. Let it be reviewed and corrected.

5. It is not stated in the Materials and Methods section where the cell lines and viruses were purchased or provided. Let it be added.

Author Response

In the manuscript submitted to me for review entitled "NLRC3 attenuates antiviral innate immune response by targeting IRF7 in grass carp (Ctenopharyngodon idelus)“ the authors Lei Zhang, Hai-tai Chen, Xiang Zhao, You-cheng Chen, Shen-peng Li, Tiao-yi Xiao and Shu-ting Xiong present a study in which they provide a new theoretical understanding of the mechanism of immune regulation in grass carp fish.

The study was conducted in vitro and in vivo, and the experiments with fish were approved by the Animal Care Committee of Hunan Agricultural University, China.

My remarks and recommendations to the authors are:

  1. The Materials and Methodssection does not describe how many experimental groups the fish used (grass carp) were divided into and how many specimens were included in each group.

Response: we have added it in the Materials and Methods section 4.3 (lines 380-387)

  1. On lines 384-385 it is stated:

"At the same time, CIK cells were seeded in the six-well plate and treated with GCRV-JX0901." Was the treatment performed immediately after the cells were seeded or after the formation of the cell monolayer?

Response: Thanks for your comments. We have revised the description revised them in blue “At the same time, CIK cells were seeded overnight in the six-well plate and then treated with GCRV-JX0901 when the cell density is approximately 5*10^5 cells/mL.”(line 389-390).

  1. In the same sentence on lines 384-385, the density of the cells that were seeded is not stated. Let it be added.

The same remark applies to sections 4.6. and 4.7.

Response: Thanks for your comments. We have revised the description of the experimental treatment in more detail and revised them in blue (line 389-390) (line 413) (line 422-423) (line 439-440).

  1. In section 4.7. in several places the grammatically correct tense is not used to describe the conduct of the experiment. Let it be reviewed and corrected.

Response: we have revised them in blue (line 429-430, 432, 434, 436-437).

  1. It is not stated in the Materials and Methodssection where the cell lines and viruses were purchased or provided. Let it be added.

Response: Thanks for your comments. We have added the source of the cell lines and viruses. “The GCRV JX-0901 strain (used in cell lines) induces significant cytopathic effects in CIK cell lines, and the GCRV-Huan1307 strain (used in individuals) causes severe hemorrhage symptoms in individuals. All the cell lines and viruses used in this experiment were long-term kept in our laboratory.” (line 363-367).